# Flexural Behavior of a Precast Concrete Deck Connected with Headed GFRP Rebars and UHPC

**DOI:** 10.3390/ma13030604

**Published:** 2020-01-29

**Authors:** Won Jong Chin, Young Hwan Park, Jeong-Rae Cho, Jin-Young Lee, Young-Soo Yoon

**Affiliations:** 1School of Civil, Environmental, and Architectural Engineering, Korea University, Seoul 02841, Korea; wjchin@kict.re.kr; 2Department of Infrastructure Safety Research, Korea Institute of Civil Engineering and Building Technology, Goyang 10223, Korea; yhpark@kict.re.kr (Y.H.P.); chojr@kict.re.kr (J.-R.C.); 3School of Agricultural Civil & Bio-industrial Engineering, Kyungpook National University, Daegu 41566, Korea

**Keywords:** glass fiber-reinforced polymer (GFRP) rebar, ultra-high-performance concrete (UHPC), concrete headed GFRP rebar, bond strength, development length, flexural strength, precast concrete deck

## Abstract

Steel bent reinforcing bars (rebars) are widely used to provide adequate anchorage. Bent fiber-reinforced polymer (FRP) rebars are rarely used because of the difficulty faced during the bending process of the FRP rebars at the construction site. Additionally, the bending process may cause a significant decrease in the structural performance of the FRP rebars. Therefore, to overcome these drawbacks, a headed glass fiber-reinforced polymer (GFRP) rebar was developed in this study. The pull-out tests of the headed GFRP rebars with diameters of 16 and 19 mm were conducted to evaluate their bond properties in various cementitious materials. Moreover, structural flexural tests were conducted on seven precast concrete decks connected with the headed GFRP rebars and various cementitious fillers to estimate the flexural behavior of the connected decks. The results demonstrate that the concrete decks connected with the headed GFRP rebar and ultra-high-performance concrete (UHPC) exhibited improved flexural performance.

## 1. Introduction

### 1.1. General

The corrosion of reinforcing steel in structures decreases its life expectancy and causes extensive maintenance costs. In this context, fiber-reinforced polymer (FRP) reinforcing bars (rebars) have been extensively used as an alternative to steel reinforcement owing to their non-corrosive characteristics. Moreover, FRP has additional advantages as a construction material, such as high specific strength, relatively lightweight, and non-conductivity. However, the substitution of a steel rebar with an FRP rebar has several difficulties owing to the differences between them. While steel bent rebars are used to provide sufficient anchorage, bent FRP rebars are rarely used because the bending process of FRP rebars at construction sites is difficult. Additionally, the bending process can cause a significant decrease in the structural performance of FRP rebars. These problems have resulted in the development of FRP bars with a headed end to satisfy the required development length. Therefore, the Korea Institute of Civil engineering and Building Technology (KICT) developed new headed FRP rebars to improve their practical applications.

Although several researchers have studied the bond strength of FRP rebars, a comprehensive relationship between the strength of concrete and FRP rebars has not been established because various companies manufacture FRP rebars using their own distinct methods. In other words, FRP rebars have not been standardized yet. Therefore, if a new type of FRP rebar is developed, the mechanical properties and bond characteristics need be evaluated as the bond characteristics of FRP rebars are important factors that govern the design of FRP reinforced structural members.

Thus, in this study, newly developed glass fiber-reinforced polymer (GFRP) rebars and headed GFRP rebars are introduced. A total of 44 specimens were fabricated and tested to estimate the pull-out behavior of GFRP rebars considering various aspects, such as the diameter of the rebars, the strength of concrete, and the types of headed end. Moreover, flexural tests were conducted on seven precast concrete decks connected with headed GFRP rebars and various cementitious fillers to determine their flexural behavior.

### 1.2. Manufacturing of a GFRP Rebar

The easiest way to increase the tensile strength of a GFRP rebar is to use high-performance constituent materials and reduce the amount of materials to overcome the problem of a price increase. To achieve superiority over ordinary rebars, the structural performance of the GFRP rebar was enhanced along with price reduction through efficient manufacturing. In this study, e-glass fiber and vinylester resin were used to manufacture an optimized GFRP rebar while maintaining a balance between cost and strength properties. A modified braidtrusion process, which is a combination of braiding and pultrusion, was applied to develop the GFRP rebar, as illustrated in Figure 1. The modified braidtrusion process was used to impart special features by pre-tensioning the glass fibers. This technique further includes a strand for forming ribs on the surface of the rebar using braiding fibers to obtain better bonding through a single process. The tightening of fibers to form the ribs results in the reduction of voids in the cross-section and increases the tensile strength of the GFRP rebar up to 900 MPa through the pre-tensioning of the core fiber bundle, as depicted in Figure 2. Figure 3 depicts the shape of the efficiently manufactured GFRP rebar in this study [1].

### 1.3. Concrete Headed GFRP Rebar

In this study, a new concrete-headed GFRP rebar is proposed, in which the head is formed by cutting the end of the GFRP rebar in the longitudinal direction and casting a concrete-like filler (mortar, ultra-high-performance concrete (UHPC), etc.) (Figure 4). Our aim is to increase the interfacial area and reduce the adhesive length. In comparison with the plastic headed GFRP rebar [2], it has a lower production cost and can be integrated with concrete. Our objective is to achieve a pullout strength capacity greater than 60% of the maximum strength of the GFRP rebar with a minimal head length. The proposed concrete-headed GFRP rebar causes bond failure when the concrete head length is short and causes FRP tensile failure when the head length is long. However, the tensile strength is lower than that of ordinary GFRP bars owing to the cross-sectional loss caused by longitudinal cutting (Figure 5) [3,4,5].

## 2. Experimental Program

### 2.1. Pull-out Test of GFRP Rebars

#### 2.1.1. Material Properties and Test Specimens

To evaluate the pull-out behavior of the normal GFRP and headed GFRP rebars, 44 specimens were fabricated and tested in this experimental study. As presented in Table 1, the parameters considered in this study are the compressive strength of concrete: the normal strength concrete (27, 35 MPa), high-strength mortar (83 MPa), and ultra-high-performance concrete (165 MPa), and the type of head: longitudinal cut, concrete head, and bar diameters: 16 mm, 19 mm.

The tensile strength of the GFRP rebars was estimated by the direct tensile test according to ACI 440.1R standard [6]. The mechanical properties of the GFRP rebars used in this study are shown in Figure 6 and Table 2. The tensile strength of the rebars used was 1050 MPa, approximately.

The concrete blocks were cast with normal strength concrete, high strength mortar, and ultra-high-performance concrete (UHPC). The compressive strengths at 28 days were measured by three concrete cylinders for each mix in accordance with ASTM C39 [7], and the results are presented in Table 1. The mix proportions of these concretes and mortars are presented in Table 3.

In accordance with ACI 440.1R [6], the GFRP rebar was embedded in a 200 mm cubic block. Two types of heads were considered in this study. The first one was a longitudinal cut head (LC), which was cut and embedded directly in the cube. Another one was a concrete head (CH), which had a concrete head, as shown in Figure 4a. The concrete heads were made of high strength mortar. To avoid rupturing at the gripping part of the GFRP rebar, the steel jackets covered a length of 700 and 1000 mm on the 16 and 19 mm diameter bar specimens, respectively, because the traditional wedge-shaped frictional grip can cause damage at the gripping part and the FRP rebar is vulnerable to shear force. The details of the specimens are depicted in Figure 7. Ten straight GFRP rebars (D16) without any type of head were prepared and tested to evaluate the bond strength of the GFRP rebar as a preliminary test. These specimens have an embedment length of 80 mm (5d) according to the ACI 440.1R standard. Additionally, a total of 34 headed GFRP rebars were fabricated and tested. Sixteen specimens were made of GFRP rebars with the longitudinal cut head and normal concrete (27, 35 MPa). Eighteen specimens were made of GFRP rebars with the concrete head and three types of concrete blocks (NC, HM, UH).

#### 2.1.2. Testing Method and Setup

The pull-out test setup is illustrated in Figure 8. The tests were performed using a universal testing machine with a capacity of 1000 kN. LVDTs were installed to measure the pull-out displacement: A, as depicted in Figure 8. The load was applied to the GFRP rebar with a rate of 0.02 mm/s.

### 2.2. Flexural Test of a Connected Precast Concrete Deck

#### 2.2.1. Material Properties and Test Specimens

Many researchers have focused on improving the precast concrete deck connections [2,8,9,10,11]. To decrease the connected length and improve flexural performance, headed GFRP rebars were developed. The flexural test on the connected precast concrete decks using two types of headed GFRP bars were conducted to verify the performance and practical effectiveness of the developed headed GFRP rebars. As depicted in Figure 9, using the headed GFRP rebar can decrease the development length in comparison with the ordinary splice connection. The length of the ordinary splice connection and that of the connection with the headed GFRP rebar were calculated according to Equations (1) and (2) [12]. The tensile strength $f_{fuh}$ of the headed FRP bar can be expressed as the product of the effective strength factor $\gamma_{h}$ of the headed bar and the tensile strength of the FRP bar, as shown in Equation (3) [12].
(1)$$l_{d}=\;\frac{\alpha \frac{f_{fr}}{0.083\sqrt{f_{ck}}}\;\;-340\;}{13.6+\;\frac{C}{d_{b}}}\;d_{b}$$
(2)$$l_{d}=\;\frac{\alpha \frac{f_{fr}-f_{fuh}}{0.083\sqrt{f_{ck}}}\;\;-340\;}{13.6+\;\frac{C}{d_{b}}}\;d_{b}$$
(3)$$f_{fuh}=\;\gamma_{h}f_{fu}$$

Here, $f_{fr}$= the required strength of FRP reinforcement; $f_{fuh}$= the tensile strength of the headed FRP bar; α = the rebar-position modification factor; C = the smallest value of cover thickness from the centroid of rebar and 1/2 of the central spacing of rebar; $d_{b}$ = the nominal diameter of rebar.

A total of seven specimens of size 240 × 450 × 2500 mm were fabricated in this study. Table 4 lists the test variables. One non-connected specimen was fabricated using straight GFRP rebars and normal strength concrete as a reference specimen. Two head types, LC and CH, filled in the connected zone, and normal strength concrete, high strength mortar, and UHPC were considered as test variables. Figure 10 depicts the details of specimens, and Figure 11 depicts the photograph of the specimens.

#### 2.2.2. Testing Method and Setup

All specimens were tested with a four-point loading condition, as depicted in Figure 12. The length between the loading points is 600 mm. The decks are simply supported with a span length of 2300 mm. All specimens were loaded monotonically until failure. This loading configuration resulted in the connection region being subjected to a constant moment without shear. The vertical displacements were measured by using LVDTs at the loading points and the center of the deck (Figure 13b).

## 3. Test Results

### 3.1. Pull-Out Properties of GFRP Rebars

As stated in ACI 440.1R, pull-out and splitting are two dominant failure modes expected with GFRP rebars in concrete [6]. For D16-NC35-N, all specimens failed in the pullout failure mode. The bond strength–slip curves of the representative specimens are illustrated in Figure 14. The result of the pull-out test of these specimens is also summarized in Table 5. In this test, the average bond strength was 10.84 MPa and the average end slip was 1.66 mm.

Figure 15, Figure 16 and Figure 17 depict the comparison of failure strengths among the specimens, and Table 6 summarizes the test results of the specimens made of concrete-headed GFRP rebars. On analyzing the data, a few of the data exhibiting inconsistencies were excluded. Consequently, the specimens made with LC and CH failed at a similar failure load. The D16 series exhibited a higher failure load than that of the D19 series. This is consistent with the observations of the previous studies [4,10], which demonstrated that as the diameter of the rebar increased, the bond strength decreased. Even the compressive strength of HM was much higher than the strength of NC, and the lower failure loads were measured in the HM specimens. It is important to note that the 200 mm cube made of UHPC exhibited a significantly higher failure load in comparison with the other specimens. Consequently, the improved structural performance can be expected when UHPC is used for the filler material that is cast in the connection of the precast deck.

### 3.2. Flexural Strength of Connected Precast Concrete Deck

All the specimens were loaded monotonically for failure to estimate the ultimate flexural capacity of each connection type. Figure 18 illustrates the load–displacement curves of the connected precast concrete deck. Figure 18a depicts the impropriety of the connection that was connected with the spliced straight GFRP rebar and high strength mortar. Similarly, the specimens made of the concrete headed GFRP rebar and high-strength mortar failed under a significantly low load of less than 100 kN, as depicted in Figure 18c. Only the specimen made with the LC GFRP rebar or GLC-HM exhibited similar behavior to the reference specimen among the specimens connected with the high-strength mortar.

In Figure 18b,d,f, all the specimens using UHPC exhibited significantly excellent flexural performance. These experimental results indicate that the use of UHPC as a filler at the connection is appropriate. From the results, the length of the connection in the precast concrete deck can be decreased when the headed GFRP rebars are used, except in the case of GCH-HM. Figure 19 depicts the flexural behavior of GSR-R (Reference) and GLC-UH.

## 4. Conclusions

The pull-out behavior of GFRP rebars in concrete was estimated considering the rebar diameter, concrete strength, and head type. Additionally, flexural tests were conducted on the precast concrete decks that were connected with various types of GFRP rebars. From the above discussions, the following conclusions are drawn:In this study, two types of headed GFRP rebars were developed and tested, and their basic pull-out mechanical properties were evaluated. Additionally, by conducting the flexural test on the connected precast concrete deck, the practical effectiveness of the headed GFRP rebar was confirmed.The pull-out test results confirmed the tendency of rebars with larger diameters to have a lower failure strength. In particular, it is important to note that the specimen made with UHPC exhibited a significantly higher failure load in comparison with the other specimens. Consequently, improved structural performance can be expected when UHPC is used as the filler material at the connection between precast deck slabs.The length of the connection in the precast concrete deck can be decreased with the use of headed GFRP rebars. The flexural test results verified that the headed GFRP rebars can provide effective anchorage performance. The LC and CH types did not exhibit significant differences in the structural performance at the connection zone of the precast decks. It is interesting to note that all the specimens using UHPC exhibited excellent flexural performance.

## Figures and Tables

**Figure 1 materials-13-00604-f001:**
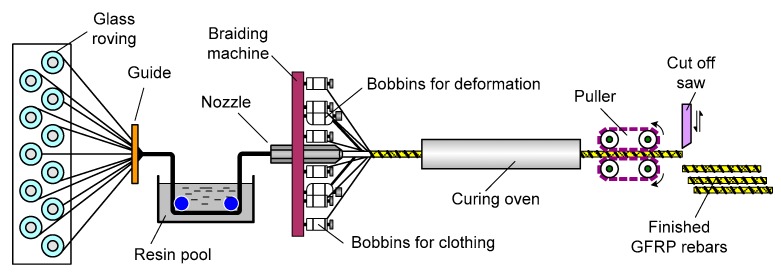
Modified braidtrusion process.

**Figure 2 materials-13-00604-f002:**
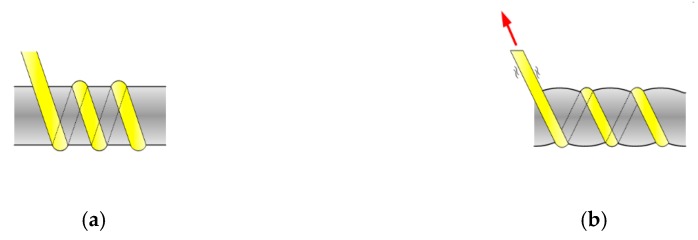
Schematic depicting the manufacturing of glass fiber-reinforced polymer (GFRP) reinforcing bars (rebars). (**a**) general braiding. (**b**) modified braiding.

**Figure 3 materials-13-00604-f003:**
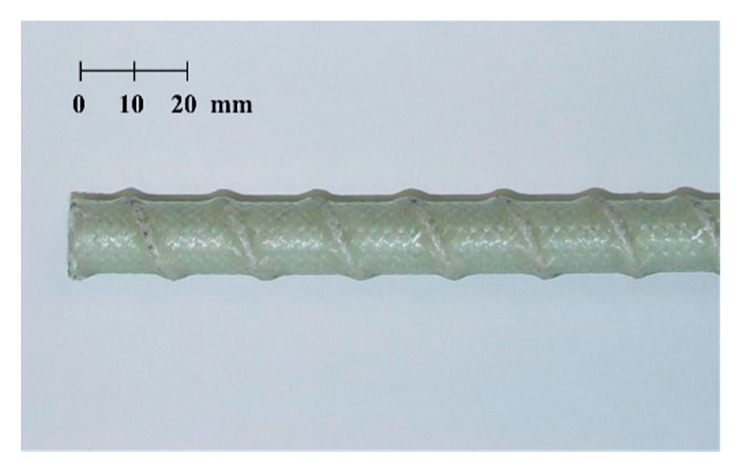
Shape of the Korea Institute of Civil engineering and Building Technology-(KICT) developed GFRP rebar.

**Figure 4 materials-13-00604-f004:**
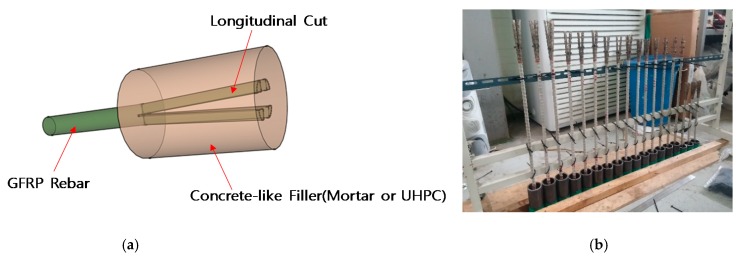
Newly developed concrete headed GFRP rebar. (**a**) Conceptual diagram. (**b**) Photograph of concrete headed GFRP rebar.

**Figure 5 materials-13-00604-f005:**
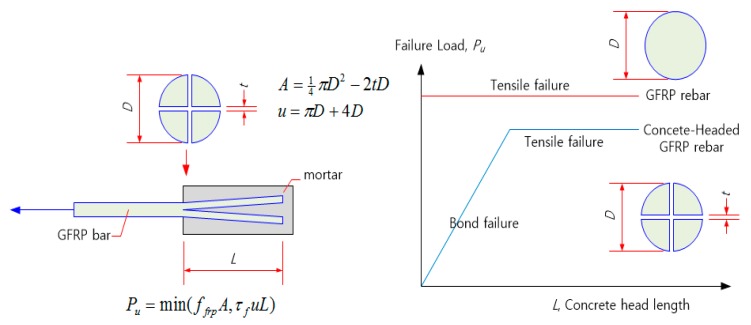
Schematic diagram of the failure load and failure mode.

**Figure 6 materials-13-00604-f006:**
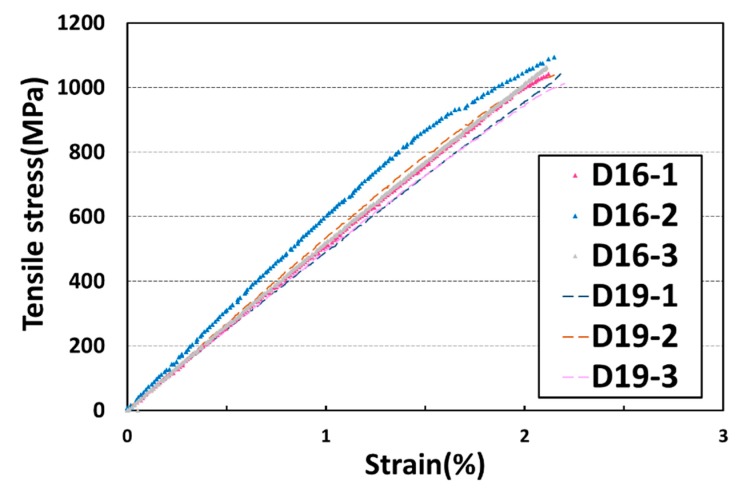
Tensile strength–strain curves of GFRP rebars.

**Figure 7 materials-13-00604-f007:**
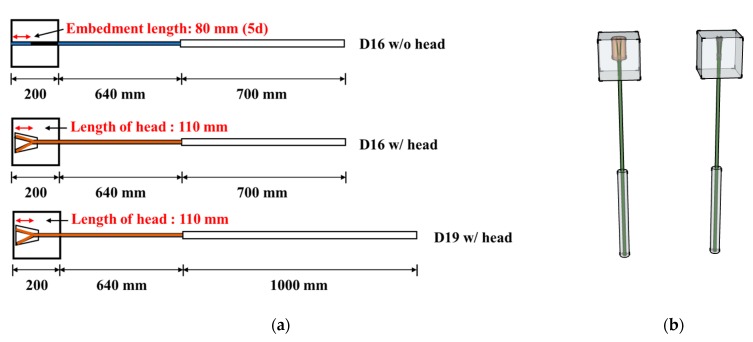
Details of specimens for the pull-out test: (**a**) specimen profiles and (**b**) a schematic view of the longitudinal cut head (LC) and concrete head (CH) specimens.

**Figure 8 materials-13-00604-f008:**
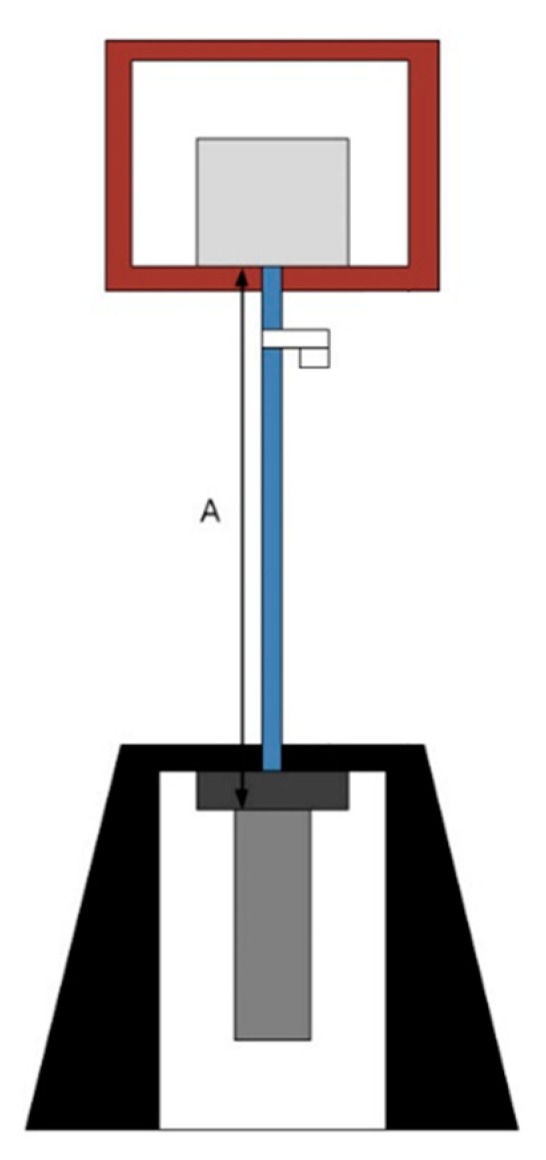
Test setup.

**Figure 9 materials-13-00604-f009:**
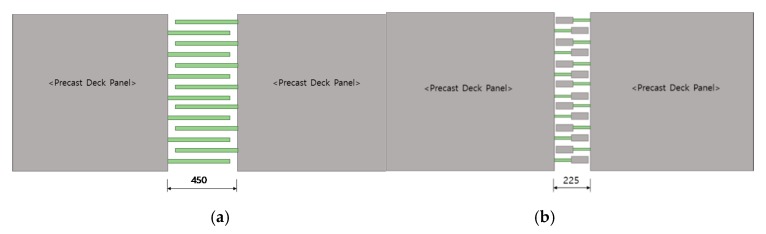
Schematic diagram of (**a**) the connection joint of the splice of GFRP rebars and (**b**) the connection joint of headed GFRP rebars.

**Figure 10 materials-13-00604-f010:**
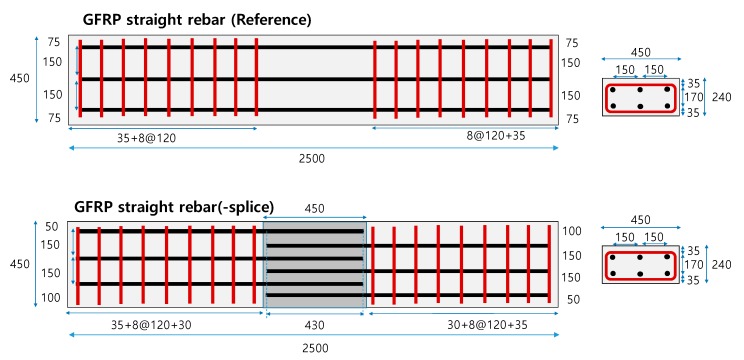
Details of the connected precast concrete deck.

**Figure 11 materials-13-00604-f011:**
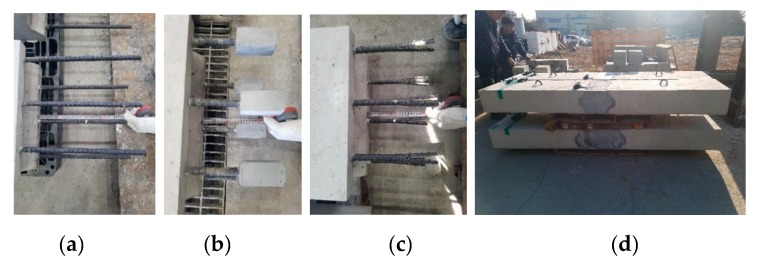
Photographs of the specimens: (**a**) a spliced straight rebar, (**b**) a concrete headed rebar, (**c**) a non-concrete headed rebar, and (**d**) a connected precast concrete deck.

**Figure 12 materials-13-00604-f012:**
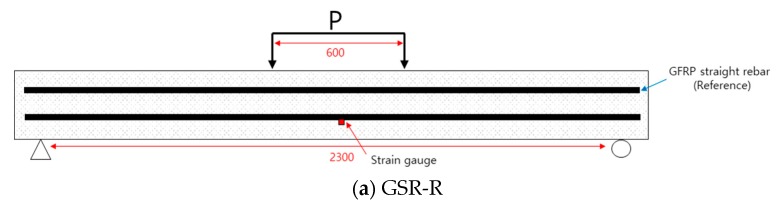
Schematic diagrams of test setup for flexural test. (**a**) GSR-R. (**b**) GSR-HM, GSR-UH. (**c**) GCH-HM, GCH-UH. (**d**) GLC-HM, GLC-UH.

**Figure 13 materials-13-00604-f013:**
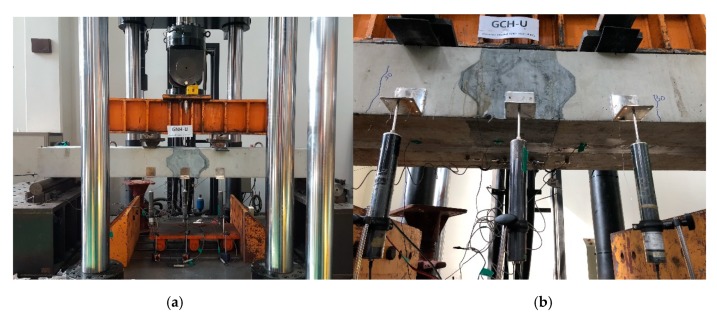
Test setup: (**a**) a photograph of the test setup and (**b**) a photograph of the measurement sensor (LVDTs).

**Figure 14 materials-13-00604-f014:**
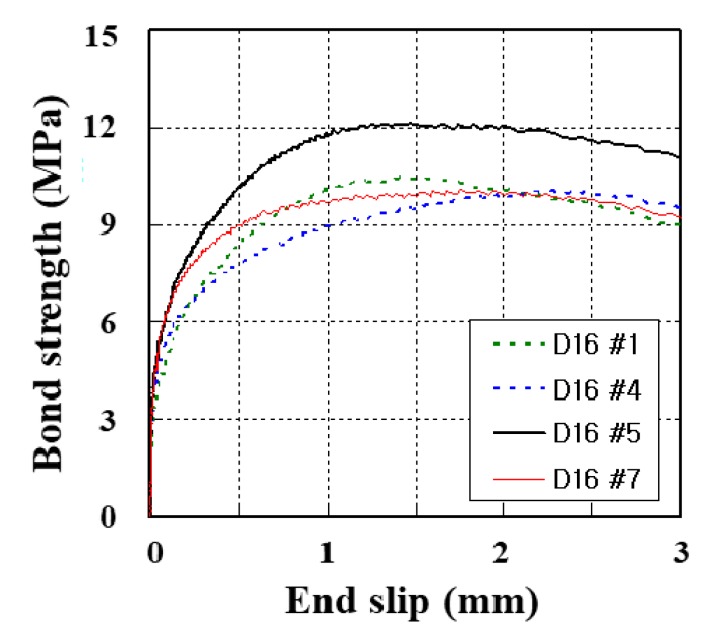
Bond strength–slip curves of GFRP rebars.

**Figure 15 materials-13-00604-f015:**
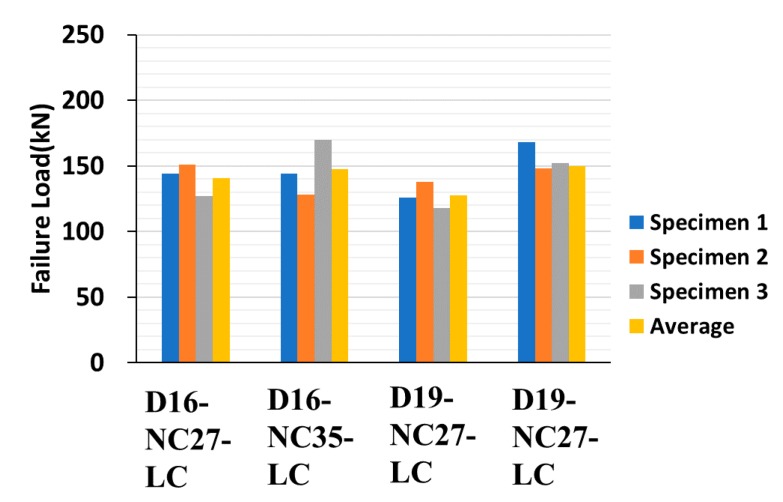
Pull-out test results of 16 and 19 mm GFRP rebars (27 MPa, 35 MPa,).

**Figure 16 materials-13-00604-f016:**
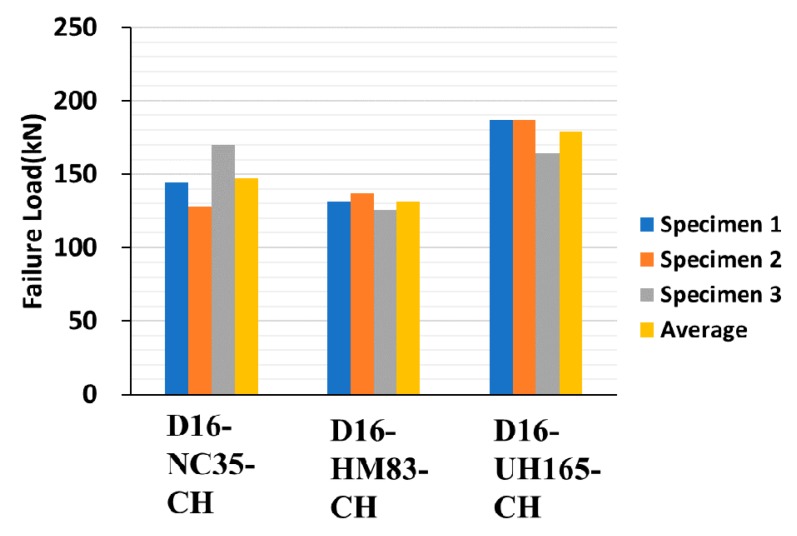
Pull-out test results of 16 mm GFRP rebars (normal concrete, mortar, UHPC).

**Figure 17 materials-13-00604-f017:**
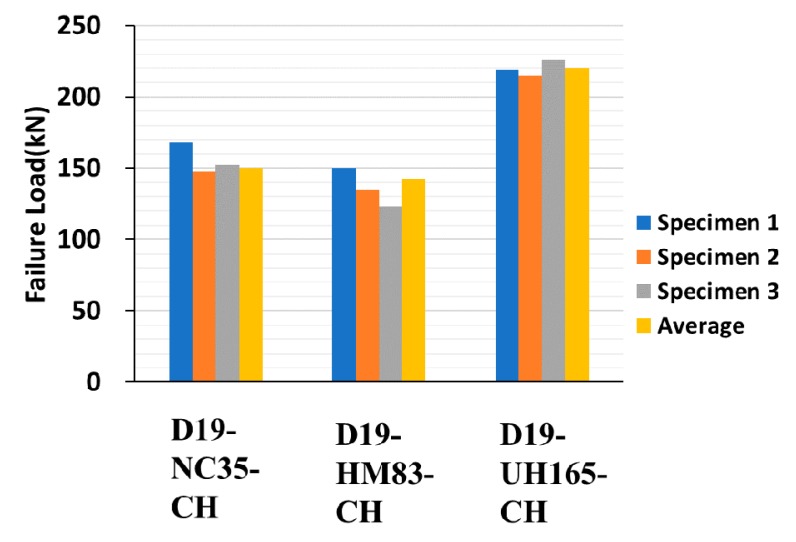
Pull-out test results of 19 mm GFRP rebars (normal concrete, mortar, UHPC).

**Figure 18 materials-13-00604-f018:**
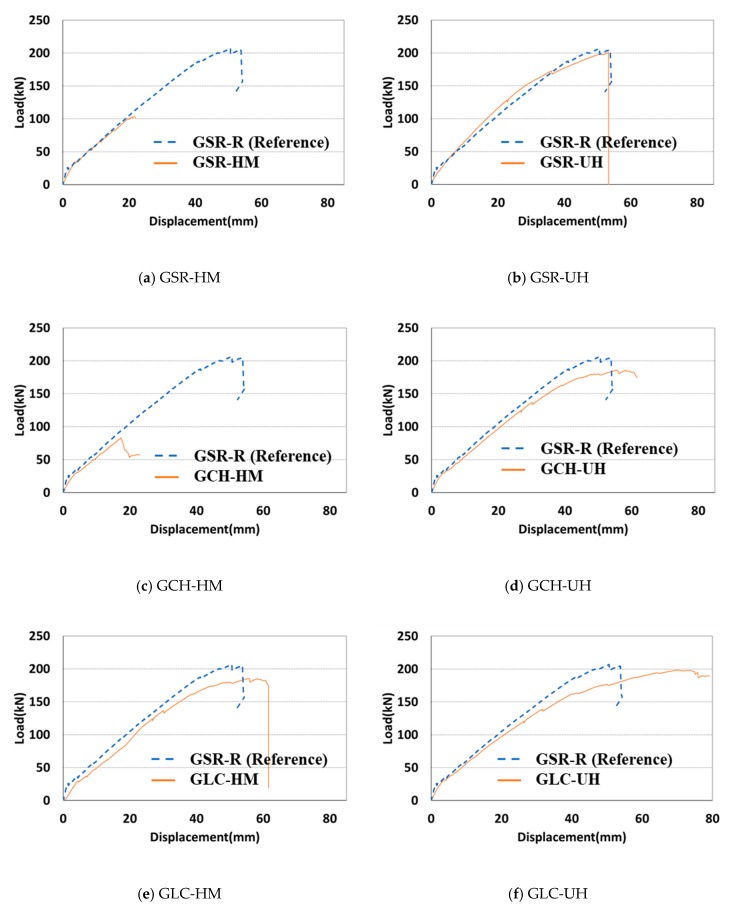
Load–displacement curves of a connected precast concrete deck. (**a**) GSR-HM. (**b**) GSR-UH. (**c**) GCH-HM. (**d**) GCH-UH. (**e**) GLC-HM. (**f**) GLC-UH

**Figure 19 materials-13-00604-f019:**
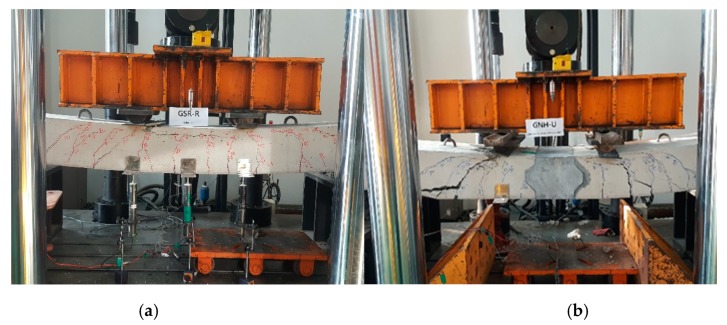
Flexural behavior of (**a**) GSR-R (Reference) and (**b**) GLC-UH.

**Table 1 materials-13-00604-t001:** Mechanical properties of test specimens.

Notation	d (mm)	C	*f_ck_* (MPa)	Type of Head	No. of Specimens
D16-NC35-N	16	NC	35.0	W/O head	10
D16-NC27-LC	16	NC	27.0	LC	4
D16-NC35-LC		NC	35.0		4
D19-NC27-LC	19	NC	27.0		4
D19-NC35-LC		NC	35.0		4
D16-NC35-CH	16	NC	35.0	CH	3
D16-HM83-CH		HM	83.2		3
D16-UH165-CH		UH	165.6		3
D19-NC35-CH	19	NC	35.0		3
D19-HM83-CH		HM	83.2		3
D19-UH165-CH		UH	165.6		3

D: diameter of FRP bar; C: Type of cementitious filler; *f_ck_*: Compressive strength of filler; NC: normal strength concrete; HM: high strength mortar; UH: ultra-high performance concrete; LC: longitudinal cut without concrete head; CH: longitudinal cut with concrete head.

**Table 2 materials-13-00604-t002:** Tensile properties of GFRP rebars.

Tensile Properties	D16	D19
Maximum tensile strength, $f_{u,ave}$, MPa	1066.39	1030.38
Nominal tensile strength, $f^{*}{}_{fu}=f_{u,ave}-3\sigma$, MPa	987.55	981.38
Design tensile strength, $f_{fu}=C_{E}f^{*}{}_{fu}(C_{E}=0.8)$, MPa	691.28	686.97
Elastic modulus, $E_{frp}$, GPa	47.84	46.71

σ: standard deviation; *C_E_*: environmental reduction factor.

**Table 3 materials-13-00604-t003:** Mix designs of specimens.

Variables	W/B (%)	Unit Weight (kg/m^3^)						
		W	C	FA	CA	AD	Steel Fiber	SP (g)
NC	40.0	226	562	598	786	0	0	-
HM	42.5	170	200	1640 *	0	200 (Fly ash)	0	5.25
UHPC	20.0	183	799.5	880	0	84 (Silica fume)	78 (19.5 mm) 39 (16.3 mm)	18.4

W/B: water to binder ratio; W: water; C: cement; FA: fine aggregate; CA: coarse aggregate; AD: admixture, SP; superplasticizer; * proportion of fine aggregates: silica sand: crushed sand = 800:840。

**Table 4 materials-13-00604-t004:** Test variables for flexural test on the connected precast concrete deck.

Specimen	Connection Joint	Connection Joint Length	Filler
GSR-R	None	Reference	Reference
GSR-HM	Spliced straight GFRP rebar	450 mm	83 MPa Mortar
GSR-UH	Spliced straight GFRP rebar	450 mm	165 MPa UHPC
GCH-HM	Concrete Headed GRFP rebar	225 mm	83 MPa Mortar
GCH-UH	Concrete Headed GRFP rebar	225 mm	165 MPa UHPC
GLC-HM	Non-concrete Headed GRFP rebar	225 mm	83 MPa Mortar
GLC-UH	Non-concrete Headed GRFP rebar	225 mm	165 MPa UHPC

**Table 5 materials-13-00604-t005:** Test results of the pull-out test (D16 GFRP rebars, normal concrete).

Specimens	Max. Load (kN)	Bond Strength (MPa)	End Slip (mm)	Failure Mode
D16-1	41.92	10.42	1.410	pullout
D16-2	42.71	10.62	1.335	pullout
D16-3	41.91	10.42	1.840	pullout
D16-4	40.49	10.07	2.265	pullout
D16-5	48.78	12.13	1.465	pullout
D16-6	40.46	10.06	2.305	pullout
D16-7d	40.45	10.06	1.755	pullout
D16-8	45.59	11.34	1.420	pullout
D16-9	45.36	11.28	1.410	pullout
D16-10	48.19	11.98	1.365	pullout
Ave.	43.58	10.84	1.66	-
STDEV	3.18	0.79	0.37	-
C.O.V.	7.3%	7.3%	22.3%	-

**Table 6 materials-13-00604-t006:** Test results of the pull-out test (D16 and D19 headed GFRP rebars, normal concrete, mortar, UHPC).

Specimen	No.	Pull Out Displacement (mm)	Max. Load (kN)
D16-NC27-CH	1	632	144
	2	644	151
	3	643	127
D16-NC35-CH	1	534	144
	2	639	128
	3	636	170
D16-HM83-CH	1	651	131
	2	654	137
	3	659	126
D16-UH165-CH	1	657	187
	2	656	187
	3	665	164
D19-NC27-CH	1	755	126
	2	753	138
	3	752	118
D19-NC35-CH	1	758	168 *
	2	754	148
	3	754	152
D19-HM83-CH	1	778	150
	2	778	135
	3	778	123 *
D19-UH165-CH	1	774	219
	2	778	215
	3	777	226

* inconsistent data excluded in mean values.

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
