# Peer review of "Flexural Behavior of a Precast Concrete Deck Connected with Headed GFRP Rebars and UHPC"

_materials, 2020, doi:10.3390/ma13030604_

Round 1

Reviewer 1 Report

Thank you for preparing the draft. Please find my comments and grammar corrections in the attached file.

I hope you will work a bit more on the discussion of the results and offer explanation for some of the results.

Author Response

Subject: Re-Submission of Manuscript for paper (620221)

Please find the attached manuscript titled “Flexural behavior of precast concrete deck connected

with headed GFRP rebars and UHPC” by Won Jong Chin, Young Hwan Park, Jeong-Rae Cho,

Jin-Young Lee, Young-Soo Yoon for possible publication as a research paper in the journal of

Materials.

This manuscript has been changed to accommodate all of the reviewer’s comments.

Please find the ‘Response to Reviewer’s Comments’ in the following pages.

I would be pleased to provide any additional information that might be required.

Thank you.

Yours truly,

Won Jong Chin

Reviewer 2 Report

Refer to attached file.

Author Response

(The authors gave the same response as above.)

Reviewer 3 Report

The research is interesting and the authors are commended on the extensive experimental work. But the manuscript needs to be revised for English grammar extensively and some of the results needs to be explained in more detail. The paper write-up is closer to a conference paper than a journal paper. In addition, no references are cited in the introduction detailing the background of the research. While correcting the English is beyond the scope of the review, a few ways to improve the manuscript presentation are listed below:

Figure 6: While all other figures are color, the figure that would have needed colored lines the most has been presented in black and white. As all the lines are bunched up, please consider using different colors along with different line types to present the stress-strain curve.  Please consider calling the connection of GSR-UH/GSR-HM as spliced straight GFRP rebar instead of GFRP straight rebar. All the rebar used in the testing is straight. So this is confusing.  Table 6: There is no information on what A, B, and C given in the table heading row is in the text. 

Author Response

First of all, thank you very much for your useful comments on our paper. We have carefully considered all your comments, and the revised manuscript is now attached for your reconsideration. We really appreciate the opportunity to resubmit. Also, we would like to thank you for your excellent comments which significantly improved the quality of our paper.

I’d like to let you know, this manuscript have been revised two times already. This is the third revision!

It seems like you reviewed the first version of the manuscript, so many things were already revised.

You can find attatched “responses to reviewers’ comments’ and you can also find out what things were changed during the first and second revisons.

- Three refererces were cited in the introduction part.

- Figure 6 was replaced with colored graph in accordance with your comment.

- In Table 6, B, C were deleted, and “A” was changed into ‘pull out displacement’ for helping readers’ understanding.

- All “GFRP straight rebar” changed into “Spliced straight GFRP rebar”

Thank you again for wonderful comments and your efforts.

Reviewer 4 Report

This study deals with the development of the headed GFRP rebar in order to overcome the drawbacks in bending process of FRP rebars. The pull-out tests of the headed GFRP rebars have been carried out to evaluate bond properties of the headed GFRP rebars in concrete. The pull-out behavior was estimated taking into account the rebar diameter, concrete strength, and head types. Moreover, the flexural test was carried out on the precast concrete decks connected with various types of GFRP rebars and various cementitious fillers to estimate the flexural behaviour of the connected decks.

From my point of view, the following aspects should be addressed by the authors before considering the manuscript for publication in the Materials Journal:

2.1.1 Material properties and specimens

Table 1 and Figure 6.   The nomenclature is not clear

LC  longitudinal cut

CH longitudinal cut and concrete head

D16-NC27-LC       Does this indicate that the bar with the longitudinal cut is embedded directly in the 200 mm cube? Is the cube made of normal concrete?

Figure 7.               Figure is not clear: The LC bars with normal concrete should be explained

2.2.2 Testing method and setup

Figure 9 b) Connection joint of headed GFRP. The connection joint length is reduced to 225 mm, but that reduction is not explained.

3.1 Pull-out properties of GFRP rebars

“Figure 15, 16 and 17 present the comparison of failure strength among the specimens and Table 6 summarize test results of the specimens made of concrete headed GFRP rebars. As a result, the specimens made with longitudinal cut head and concrete head failed at similar failure load. D16 series showed higher failure load than D19 series. It is important to note that the specimen made with UHPC showed significant higher failure load compared to the other specimens. Consequently, the improved structural performance can be expected when UHPC is used for the filler material that casted in the connection of precast deck”.

In pull-out specimens,     …for the concrete head?

3.2 Flexural strength of connected precast concrete deck

In my opinion, it would be more interesting to show the improvement produced by the different headed GFRP rebars, without modifying the properties of the concrete. These anchors theoretically improve the adherence by increasing the contact surface, and the improvement produced by the concrete filler avoids measuring the improvement due only to the headed GFRP.

If the aim of the work is to evaluate the improvement due to the headed GFRP rebars, the filler and the precast concrete deck should be in the same concrete (Normal concrete).

Author Response

(The authors gave the same response as above.)

Reviewer 5 Report

The content and covered topics are relevant, current, and interesting. The idea of using the headed-GFRP rebars is valuable and should be evaluated. However, the paper in its current form is not scientifically sound, as a few relevant aspects were not addressed or not properly addressed:

Technical issues:

 It not absolutely clear how E-glass and Vinylester makes most Efficient or economical rebars

Can GFRP rebars really be made stronger by winding helical fibers around? This seems questionable as it does not contribute to the direct load carrying mechanism. At best, it changes the failure mechanism (which is not described in the paper) What is CIKT? It is not explained in the paper. Pullout tests according to ACI440.1R usually include displacement/slip measurements at two locations. At the free-end and at the load-end. In this paper, only the load-end displacement is reported, whereas the free-end slip is more relevant. Clearly, the experiments are already completed and it appears that no data at the free-end was collected. However, this needs to be addressed and explained. Why was this choice made, and how can the researchers rely only on the load-end displacement? The displacement measurements at A,B, and C are not in agreement with standard pullout test (at the load-end). This needs to be explained because the reader will interpret the results based on standard tests. From the experimental setup, the reader can not clearly identify which part of the rebar was bonded and which part was shielded. In Figure 7, for the standard test, this can be guessed, but for the other tests (headed bars) this is not clear. Also, if the headed bars were not shielded, this needs to be addressed and the implications must be addressed (how does this correlate to the normal tests, in which the bars are shielded). If the headed bars were shielded, that needs to be identified and the correlation must be explained. The structural tests are well documented in images, but more explanation in the text could be provided. The specimen choice for the graph in Figure 14 is not clear. Why were random specimens chosen? Why not all? Also, the y-axis needs to identify which measurement this is (A,B, or C?). It also should state that it is the load-end slip. The graphs from Fig 15 through 17 are too detailed and potentially confusing. It is not necessary to plot individual results and average. These figures/results can be better compared if they are combined into  one. Only averages (with overplayed ranges) should be plotted. For Figure 18, the individual data lines could be placed in the same plot to better compare the data sets (specially because the reference data is the same in each plot). It is not clear to the reader why two types of experiments (pullout and slap/beam tests) are presented in this paper. No correlation between the topics is offered. A discussion is completely missing. The experimental program is great and the results (when presented correctly) can clarify great ideas/findings. However, this needs to be discussed in mored detail and compared to the literature. So far, this paper is a report/documentation of experiments and numbers. What value does this have to the engineering community?  The conclusions are more written like a summary and only the last bullet is a true conclusion. This may be due to the lack of discussion and can improve significantly if a discussion is included. Finally, a general comment for the authors to consider: The pullout tests and those results are great. And so are the beam tests. But specifically because no correlation between the two topics is offered. This paper seems overloaded with two topics. This reviewer believes that both of these two aspects could be much more thoroughly reviewed when presented individually. If the authors provide more details and a deeper discussion about each topic, they could be split up, to provide single focused papers. Currently, it is not clear why the two topics explained together. Besides the fact that both experimental procedures were conducted for the same rebar types, no correlation seems apparent.

On the non-technical side, this paper seems to be written by multiple people with different writing styles (that collide). Please unify. For example, at the beginning of the paper, the tenses switch from present tense to past tense and back and forth. Later in the paper, this is much better, but every now and then, it is written in present tense when it should be past tense. Suggestion to the Author is to rewrite/adjust everything to past tense (as the experiments are completed and are in the past).

Significant grammatical issues exist throughout the paper. Some sentence include "double words" or "double thoughts" (starting with a thought, followed by a similar thought in different words). The paper should be edited for English/grammar.

Author Response

(The authors gave the same response as above.)

Reviewer 6 Report

The paper is well written, well structured, and clear. The title clearly describes the contents of the paper. The abstract provides a concise and complete summary and the reference list is appropriate. I would suggest the article for publication. Before final publication, a few comments that the authors could take into account

Scale bars are missing from Fig 3, 4(b), 11, 13, and 19.

Fig 7: I would recommend choosing colors with the strong contract to represent different GFRP rebars, lines in Figure 7 are difficult to differentiate. Please revise.

Fig 8 (a) doesn’t add any value to the manuscript, I would suggest to remove this image.

Author Response

Thank you very much for your review on our work. The authors are encouraged by your positive comments. Also, we would like to thank you for your excellent comments which significantly improved the quality of our paper.

Scale bar was added in Fig. 3 according to your comment.

Figure 7 was re-drown with colored lines.

Figure 8 (a) was removed.

Thank you again for wonderful comments and your efforts.

Round 2

Reviewer 1 Report

Thank you for the revised version.

I appreciate that you have considered the comments and I think the text is clearer and the results easier to compare.

I still think it requires proof reading but those are minor issues.

Author Response

First of all, thank you very much for your useful comments on our paper.

Without your good comments, we could not make our paper better.

In addition, this manuscript has been entirely proofread by Editage (www.editage.co.kr), the English proofreading company.

Thank you again.

Reviewer 3 Report

I believe the English in the paper is only marginally acceptable for publication. But I also believe the research is of sufficient quality to accept for publication. The following minor grammar/spelling errors were detected.

Line 54: 'deck' should be 'decks'

Line 129: 'Two types of head types' should be 'Two types of heads'

Line 133: 'the steel jackets were covered with a length of 700 and 1000 mm on each specimen, respectively' in my opinion should be 'steel jackets covered a length of 700 and 1000 mm on the 16 mm and 19 mm diameter bar specimens, respectively' 

Please consider revising equations 1, 2, and 3 for clarity. They are of very poor resolution. 

Line 210: 'ACI 440.1R mentioned that pull-out and splitting are two dominant failure modes' should be 'As stated in ACI 440.1R pull-out and splitting are the two dominant failure modes'

Line 261:'was estimated with a consideration of rebar' should be 'was estimated considering rebar'

Line 275:'for the filler material that is casted in the connection of the precast deck' should be 'as the filler material at the connection between precast deck slabs'.

Author Response

Thank you very much for your useful comments which significantly improved the quality of our paper. We have carefully considered all your comments, and the revised manuscript is now attached.

All grammar/spelling errors were revised and highlighted in the manuscript.

- Line 54: 'deck' -> 'decks'

- Line 129: 'Two types of head types' -> 'Two types of heads'

- Line 133: 'the steel jackets were covered with a length of 700 and 1000 mm on each specimen, respectively' -> 'steel jackets covered a length of 700 and 1000 mm on the 16 mm and 19 mm diameter bar specimens, respectively' 

- Equations 1, 2, and 3 were revised.

- Line 210: 'ACI 440.1R mentioned that pull-out and splitting are two dominant failure modes' -> 'As stated in ACI 440.1R pull-out and splitting are the two dominant failure modes'

- Line 261:'was estimated with a consideration of rebar' -> 'was estimated considering rebar'

- Line 275:'for the filler material that is casted in the connection of the precast deck' -> 'as the filler material at the connection between precast deck slabs'.

Thank you again for wonderful comments and your efforts.

Reviewer 4 Report

3.2 Flexural strength of connected precast concrete deck
In my opinion, it would be more interesting to show the improvement produced by the different headed GFRP rebars, without modifying the properties of the concrete. These anchors theoretically improve the adherence by increasing the contact surface, and the improvement produced by the concrete filler avoids measuring the improvement due only to the headed GFRP.
If the aim of the work is to evaluate the improvement due to the headed GFRP rebars, the filler and the precast concrete deck should be in the same concrete (Normal concrete).

Author Response

Thank you for your excellent comment. Authors agree with your opinion. As you mentioned, if the test was conducted on the connection made of normal concrete, it’s better to show the improvement of GFRP head.

We’d like to explain the reason why we use HM and UHPC for connection.

In general, the connection part was constructed with higher strength materials in construction site. Authors wanted to reflect practical condition in this study. But if authors have the chance to conduct a further study, your comment must be considered.

Thank you again for wonderful comments and your efforts.

Reviewer 5 Report

The paper still contains similar issues as mentioned in the previous review process.

1) The bar graphs in the results section are unnecessary cumbersome and do not convey the message with a minimum of information. Why plot more than the average? Why not put all averages on the same graph? Why separate them?

2) figure 18; still as before (please see previous feedback)

3) Most importantly, this paper contains no discussion. As long as the paper does not include a discussion in which the results are contemplated and/or compared to findings made by others, this paper should not be published because the authors would leave it up to the reader how to interpret and place the findings.

4) Lastly, please read the feedback from the last review carefully. For example the measurement of the “free end” is still not addressed. Also, the paper does not flow well (please improve English).

Author Response

The authors wish to thank the reviewers for their time in effort in reviewing our manuscript.

1) The bar graph : Aouthors think this graph format is better and we'd like to  show this bar garph separately in the grouped sepcimen series.

2) The Figure 18 was revised.

3) and 4)

Aouthors agree with reviewer's opinion on our paper. This paper contains few discussion. But we think this paper is experimental focussed paper. Aouthors believe that to provide fundamental test results have a academic value.

If authors have the chance to conduct a further study, your comment must be considered.

In addition, this paper has been proofread entirely by "Editage", the proofreading company.

Thanks  again your good comments.

Round 3

Reviewer 4 Report

Acept in present form

Author Response

Thank you very much for your review on our work. The authors are encouraged by your positive comments.

Thank you once again.

Reviewer 5 Report

The paper continues to suffer from the same shortcomings that were detected in the first version.

This is the third version without a proper discussion or without an adequate comparison of the results to the literature. How will the reader be able to judge the context of this work without a discussion? Without a discussion, a manuscript cannot/should not be published (in my opinion).
Likewise, as mentioned in the first review, the first conclusion is not a conclusion, but instead it is a summary. It needs to be changed (as mentioned before). 
Finally, the new version still does not include an explanation why the „free end slip“ (see ASTM for pullout test for FRP rebars) was not measured. The „load end slip“ is insufficient to characterize the bond behavior.

Author Response

Please understnad authors' situation.

Five reviewers gave me positive reviews and comments during revision process.

Authors agree with reviewer's opinion, but your comment was very critical.

Entirely re-writing a paper is easier than revising this manuscript to meet your standard.

We aleady said that this paper is experimental focussed paper. Aouthors believe that to provide fundamental test results have a academic value.

If authors have the chance to conduct a further study, your comment must be considered.

Thanks again your good comments.